# Tryptophan metabolism determines outcome in tuberculous meningitis: a targeted metabolomic analysis

Edwin Ardiansyah[1,2], Julian Avila-Pacheco[3], Le Thanh Hoang Nhat[4], Sofiati Dian[1,5], Dao Nguyen Vinh[4], Hoang Thanh Hai[4], Kevin Bullock[3], Bachti Alisjahbana[1,6], Mihai G Netea[2], Riwanti Estiasari[7], Trinh Thi Bich Tram[4], Joseph Donovan[4,8,9], Dorothee Heemskerk[10], Tran Thi Hong Chau[4,11], Nguyen Duc Bang[12], Ahmad Rizal Ganiem[1,5], Rovina Ruslami[1,13], Valerie ACM Koeken[2,14], Raph L Hamers[15], Darma Imran[7], Kartika Maharani[7], Vinod Kumar[2], Clary B Clish[3], Reinout van Crevel[2], Guy Thwaites[4,8], Arjan van Laarhoven[2*†], Nguyen Thuy Thuong Thuong[4,8*†]

[1]Research Center for Care and Control of Infectious Diseases, Universitas Padjadjaran, Bandung, Indonesia; [2]Department of Internal Medicine and Radboud Center of Infectious Diseases (RCI), Radboud University Medical Center, Nijmegen, Netherlands; [3]The Broad Institute of MIT and Harvard, Cambridge, United States; [4]Oxford University Clinical Research Unit, Ho Chi Minh City, Viet Nam; [5]Department of Neurology, Hasan Sadikin Hospital, Faculty of Medicine, Universitas Padjadjaran, Bandung, Indonesia; [6]Department of Internal Medicine, Hasan Sadikin Hospital, Faculty of Medicine, Universitas Padjadjaran, Bandung, Indonesia; [7]Department of Neurology, Cipto Mangunkusumo Hospital, Faculty of Medicine Universitas Indonesia, Jakarta, Indonesia; [8]Centre for Tropical Medicine and Global Health, Nuffield Department of Medicine, University of Oxford, Oxford, United Kingdom; [9]London School of Hygiene and Tropical Medicine, London, United Kingdom; [10]Department of Medical Microbiology and Infection Prevention, Amsterdam University Medical Centre, Amsterdam, Netherlands; [11]Hospital for Tropical Diseases, Ho Chi Minh City, Viet Nam; [12]Pham Ngoc Thach Hospital for Tuberculosis and Lung Disease, Ho Chi Minh City, Viet Nam; [13]Department of Biomedical Science, Faculty of Medicine, Universitas Padjadjaran, Bandung, Indonesia; [14]Department of Computational Biology for Individualised Infection Medicine, Centre for Individualised Infection Medicine (CiiM) & TWINCORE, joint ventures between the Helmholtz-Centre for Infection Research (HZI) and the Hannover Medical School (MHH), Hanover, Germany; [15]Oxford University Clinical Research Unit Indonesia, Faculty of Medicine Universitas Indonesia, Jakarta, Indonesia

*For correspondence:
Arjan.vanLaarhoven@
radboudumc.nl (AL);
thuongntt@oucru.org (NTTT)

†These authors contributed equally to this work

## Abstract

**Background:** Cellular metabolism is critical for the host immune function against pathogens, and metabolomic analysis may help understand the characteristic immunopathology of tuberculosis. We performed targeted metabolomic analyses in a large cohort of patients with tuberculous meningitis (TBM), the most severe manifestation of tuberculosis, focusing on tryptophan metabolism.

**Methods:** We studied 1069 Indonesian and Vietnamese adults with TBM (26.6% HIV-positive), 54 non-infectious controls, 50 with bacterial meningitis, and 60 with cryptococcal meningitis. Tryptophan and downstream metabolites were measured in cerebrospinal fluid (CSF) and plasma using

targeted liquid chromatography–mass spectrometry. Individual metabolite levels were associated with survival, clinical parameters, CSF bacterial load and 92 CSF inflammatory proteins.

**Results:** CSF tryptophan was associated with 60-day mortality from TBM (hazard ratio [HR] = 1.16, 95% confidence interval [CI] = 1.10–1.24, for each doubling in CSF tryptophan) both in HIV-negative and -positive patients. CSF tryptophan concentrations did not correlate with CSF bacterial load nor CSF inflammation but were negatively correlated with CSF interferon-gamma concentrations. Unlike tryptophan, CSF concentrations of an intercorrelating cluster of downstream kynurenine metabolites did not predict mortality. These CSF kynurenine metabolites did however correlate with CSF inflammation and markers of blood–CSF leakage, and plasma kynurenine predicted death (HR 1.54, 95% CI = 1.22–1.93). These findings were mostly specific for TBM, although high CSF tryptophan was also associated with mortality from cryptococcal meningitis.

**Conclusions:** TBM patients with a high baseline CSF tryptophan or high systemic (plasma) kynurenine are at increased risk of death. These findings may reveal new targets for host-directed therapy.

**Funding:** This study was supported by National Institutes of Health (R01AI145781) and the Wellcome Trust (110179/Z/15/Z and 206724/Z/17/Z).

## Editor's evaluation

This important study by Ardiansyah and colleagues reports the association of tryptophan levels in cerebrospinal fluid with 60-day mortality in patients with tuberculosis meningitis. Good evidence is presented that cerebrospinal fluid tryptophan levels are associated with mortality. The findings continue to remain an association, without clarity of whether tryptophan is a key mediator of mortality or another inflammatory biomarker. The work will be of interest to tuberculosis researchers.

## Introduction

Tuberculous meningitis (TBM) is the most severe manifestation of tuberculosis affecting approximately 160,000 adults each year (*Dodd et al., 2021*). Patients suffer from varying degrees of intracerebral inflammation, commonly manifest as leptomeningitis, vasculitis, and space-occupying brain lesions (tuberculomas). Hydrocephalus, stroke, seizures, focal neurological deficits, and loss of consciousness are common complications and lead to death in around 30% of patients, even when treated with anti-tuberculosis drugs and adjuvant corticosteroid therapy (*Dodd et al., 2021*). Development of more effective host-directed therapy is hampered by a lack of knowledge on the biological pathways involved in the immunopathology of TBM (*Wilkinson et al., 2017*).

Metabolism is critical for the function of immune cells, and analysis of cerebrospinal fluid (CSF) metabolites could help unravel underlying biological mechanisms in TBM. Previously, using a large-scale metabolomics analysis, we found that lower CSF tryptophan concentrations were associated with survival of TBM patients in Indonesia (*van Laarhoven et al., 2018*). This study did not include HIV-infected patients and the association was not validated in other populations. Moreover, there is a need to investigate the downstream metabolites in the kynurenine pathway (*Figure 1*), through which 95% of tryptophan is initially catabolized via indoleamine 2,3-dioxygenase (IDO) or tryptophan 2,3-dioxygenase (TDO) and which includes metabolites with putative neuroprotective (e.g. kynurenic acid) or neurodamaging (e.g. quinolinic acid) properties (*Lovelace et al., 2017*). Lastly, there is a need to compare these findings in other neuro-infectious diseases to distinguish disease specific from broader mechanisms.

We therefore sought to define and validate the relationship between tryptophan and its metabolites and survival from TBM in large, independent populations, including HIV-positive individuals. We aimed to confirm that a higher CSF tryptophan would predict higher mortality across different populations and we hypothesized that high tryptophan would be associated with a higher CSF bacterial load, more inflammation, and lower downstream kynurenine metabolites. We lastly sought to investigate how systemic (plasma) metabolite concentrations linked to outcome.

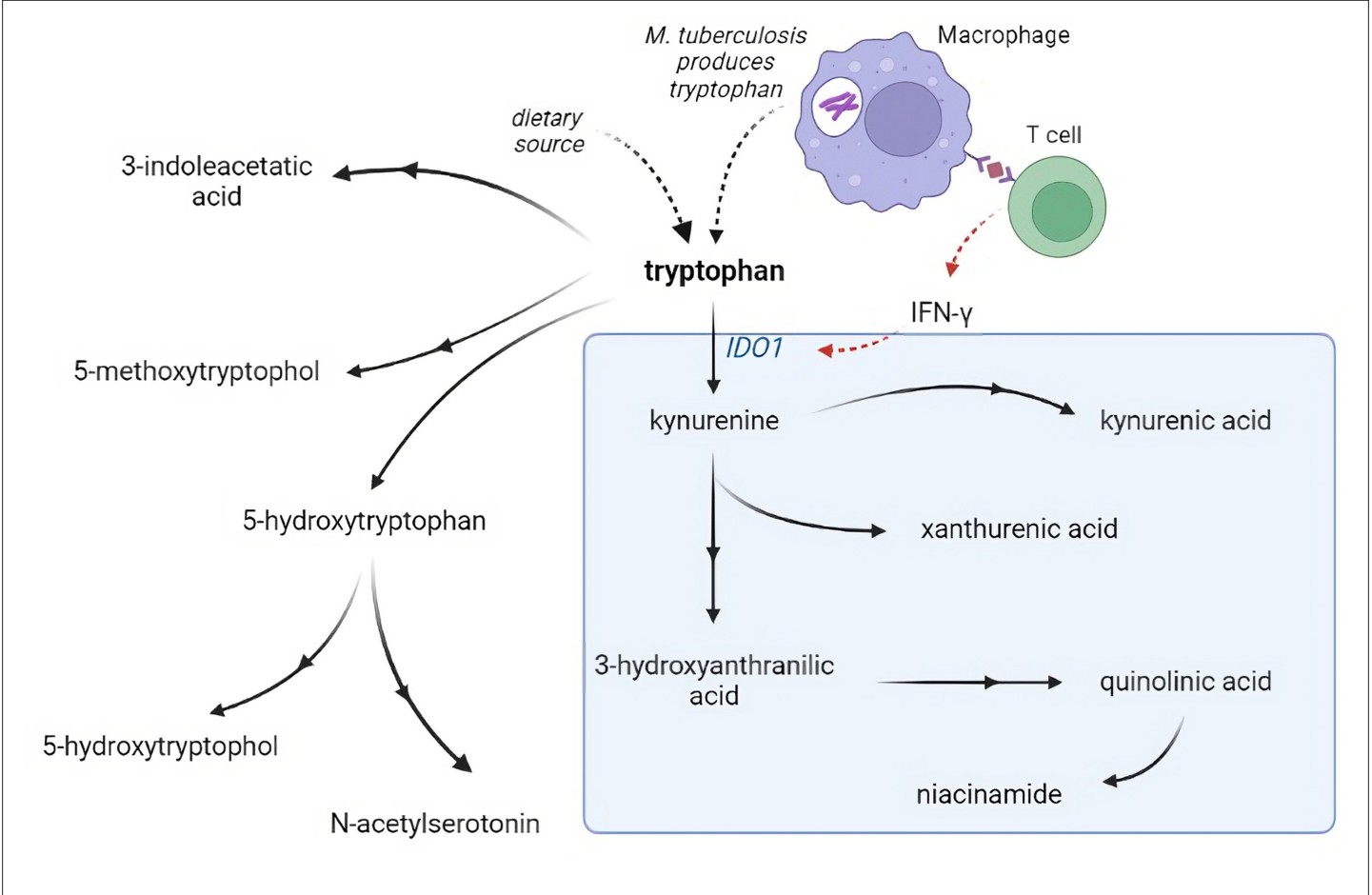

**Figure 1.** Tryptophan metabolism pathway. Tryptophan is metabolized mainly through the kynurenine pathway through indoleamine 2,3-dioxygenase 1 (IDO1), generating kynurenine and its downstream metabolites (blue box). IDO1 is partly stimulated by *M. tuberculosis*-induced interferon gamma (IFN-γ) production by T helper 1cells.

## Materials and methods

### Setting and patients

Patients with subacute meningitis were included from the Hospital for Tropical Diseases and Pham Ngoc Thach Hospital for Tuberculosis and Lung Disease in Ho Chi Minh City, Vietnam between 2011 and 2014 (*Thuong et al., 2017*; *Heemskerk et al., 2016*), and Hasan Sadikin Hospital in Indonesia between 2007 and 2019 (*van Laarhoven et al., 2018*; *van Laarhoven et al., 2017*). TBM patients were defined as having 'definite TBM' if they had either microbial confirmation by Ziehl–Neelsen staining, positive CSF culture, or GeneXpert. Based on previous studies (*van Laarhoven et al., 2017*), probable TBM was defined as clinically suspected TBM fulfilling at least two out of the following three criteria: CSF leukocytes ≥5 cells/µl, CSF/blood glucose ratio <0.5, and CSF protein >0.45 g/l. Patients were treated with antibiotics according to national guidelines for 180 days minimally, and received adjunctive dexamethasone starting at 0·3 mg/kg for grade I and 0·4 mg/kg for grade II or III TBM and tapered thereafter (*Thwaites et al., 2004*). Patients were followed-up clinically or by phone up until day 180 from admission. Primary outcome was 60-day survival, when most deaths attributable to TBM occur. As a secondary endpoint, earlier and later mortality were explored separately. We ensured equal power for both time windows by separating them by the median time to death for those patients who died during the total follow-up of 180 days. The median time to death was 14 days and was used as the cut-off to differentiate early (days 0–14) from late (days 14–180) mortality. Patients without an infection (non-infectious controls) were included from the same sites. In Indonesia, patients in this group had undergone a lumbar puncture for suspected central nervous system

infection or subarachnoid bleeding, but infection was excluded by negative microscopy, GeneXpert and bacterial culture, and CSF leucocytes <5 cells/µl and CSF/blood glucose ratio ≥0.5. In Vietnam, patients were included as controls if they had undergone a lumbar puncture, but an alternative, non-infectious, diagnosis was confirmed. In both sites, none of the non-infectious controls received anti-tuberculosis treatment. HIV-negative patients with microbiologically confirmed bacterial meningitis and HIV-positive patients with cryptococcal meningitis patients were included from the same sites.

Ethical approval was obtained from the Ethical Committee of Hasan Sadikin Hospital, Faculty of Medicine, Universitas Padjadjaran, Bandung, Indonesia (No. 114/FKUP-RSHS/KEPK/Kep./EC/2007 and No. 330/UN6.C1.3.2/KEPK/PN/2016) and from the Oxford Tropical Research Ethics Committee in the United Kingdom (OXTREC reference number: 33-09), the Institutional Review Boards of the Hospital for Tropical Diseases and Pham Ngoc Thach Hospital in Vietnam. Written (Vietnam) or oral (Indonesia) consent to be included in the study, for storage of surplus sample, and to obtain follow-up data, was obtained from patients or close relatives of patients who were unconscious. The paper adheres to the STROBE methodology.

## Metabolite measurements

CSF and blood samples were centrifuged for 15 min according to local protocols (865–3000 × $g$) and supernatants were stored at −80°C (*Rohlwink et al., 2019*). CSF and plasma metabolites were measured using targeted a liquid chromatography–tandem mass spectrometry (LC–MS) method with a system comprised of a 1290 Infinity II U-HPLC coupled to an Agilent 6495 Triple Quadrupole mass spectrometer (Agilent Tech., Santa Clara, CA). Metabolites were extracted from plasma or CSF (10 µl) using 90 µl of acetonitrile/methanol/formic acid (74.9:24.9:0.2, vol/vol/vol) containing stable isotope-labeled internal standards (valine-d8, Sigma-Aldrich, St. Louis, MO; and phenylalanine-d8, Cambridge Isotope Laboratories, Andover, MA). The samples were centrifuged (10 min, 9000 × $g$, 4°C), and the supernatants were injected directly onto a 150 × 2 mm, 3 µm Atlantis HILIC column (Waters, Milford, MA). The column was eluted isocratically at a flow rate of 250 µl/min with 5% mobile phase A (10 mM ammonium formate and 0.1% formic acid in water) for 0.5 min followed by a linear gradient to 40% mobile phase B (acetonitrile with 0.1% formic acid) over 10 min. Pairs of pooled samples generated using aliquots of all samples in the study were included every 20 samples correct for MS sensitivity drift and for quality control analyses. Sample stability over the 7-year study inclusion and 4-year storage time was checked by plotting metabolite levels of definite TBM patients against storage time. Tryptophan metabolites were measured using the following multiple reaction monitoring transitions: 3-hydroxyanthranilic acid (154.1 to >136.0), 3-indoleacetic acid (176.1 to >130.1), 3-methoxyanthranilate (168.1 to >150.0), 5-hydroxyindoleacetic acid (192.1 to >146.0), 5-methoxytryptophol (192.1 to >130.0), kynurenic acid (190.1 to >144.1), kynurenine (209.1 to >94.0), tryptophan (205.1 to >187.9), *N*-acetylserotonin (219.1 to >160.0), niacinamide (123.1 to >80.1), quinolinic acid (168.0 to >149.9), and xanthurenic acid (206.1 to >132.0). Absolute concentrations were determined using external calibration curves created via serial dilution of stable isotope-labeled compounds in CSF and plasma. These compounds were sourced from Cambridge Isotope Labs: 3-indoleacetic acid-d7 (DLM-8040), anthranilic acid-$^{13}$C6 (CLM-701), 5-HIAA-$^{13}$C6 (CLM-9936), kynurenic acid-d5 (DLM-7374), L-kynurenine-d6 (DLM-7842), L-tryptophan-$^{13}$C11 (CLM-4290), and niacinamide-$^{13}$C6 (CLM-9925). Peak abundances were manually integrated using the MassHunter software provided by the LC–MS manufacturer.

## CSF mycobacterial load and inflammatory proteins

The CSF mycobacterial load was inferred qualitatively by comparing patients with negative versus positive CSF culture, and semiquantitively from the GeneXpert Ct-values as described previously (*Thuong et al., 2019*), and inferred from CSF *M. tuberculosis* culture. CSF inflammatory cytokines in 178 Indonesian HIV-negative TBM patients were measured using a multiplex proximity extension assay (Olink) in two batches. Olink uses a multiplex assay that simultaneously recognize 96 target proteins through specific paired-antibodies which are coupled with unique oligonucleotides, for quantitative PCR measurement (*Assarsson et al., 2014*). For each protein, overlapping samples from two batches were fitted in a linear regression model, where the linear components were subsequently extracted, and used as correction factors for batches normalization. In 304 Vietnamese HIV-negative patients, 10 human cytokines were measured in CSF with Luminex multiplex bead array technology

(Bio-Rad Laboratories, Hercules, CA) (*Thuong et al., 2017*). CSF total protein was used as proxy for blood–CSF barrier disruption as it showed a near-perfect correlation with the established marker CSF–serum albumin ($r^2$ = 0.98) (*Svensson et al., 2020*).

## Quality control and statistical analysis

Only metabolites and proteins with a coefficient of variation of the pooled samples <30% and<25% missing values, respectively, among TBM patients were further included in the analysis. Remaining missing metabolite values after quality control were replaced with half of the minimum measured value of the corresponding metabolite, and log2-transformed subsequently. Statistical analyses were performed in R 4.0.4 (*R Development Core Team, 2022*), using the R packages survival, tableone, dplyr, openxlsx, pheatmap, grid, and ggplot2. Correlation analyses between metabolites levels, and between metabolites levels and clinical and inflammatory parameters, were calculated using Spearman-rank correlation. The impact of baseline CSF and plasma metabolite levels on 60-day survival was tested in a Cox-regression model, adjusted for sex, age, and HIV status as covariates. The model stratified by study site as mortality is known to be higher in the Indonesian[75] than in the Vietnamese (*Heemskerk et al., 2016*) cohort. An analysis plan was made before the study; however, it was modified because of similar metabolite levels between HIV-infected and -uninfected metabolites, we decided to incorporate HIV status as covariate, rather than a stratum, to improve power. Correction for multiple testing using the Benjamini–Hochberg method was done if multiple comparisons were done in primary analysis.

## Results

### Baseline characteristics of TBM patients and controls

We studied 1069 adults with TBM, 390 from Indonesia and 679 from Vietnam (*Table 1*). Patients were young (median age 34 years), 26.6% were HIV positive, and the majority presented with a moderately severe (55.6% grade II) to severe (17.0% grade III) disease according to the international classification (*Thwaites et al., 2003*). The rate of mycobacterial confirmation was 64.1%. Sixty-day mortality, the primary endpoint in the analysis, was 21.6%. Patients who died within 180 days from admission did so after a median of 14 days. A 14-day cut-off was therefore used to distinguish early from late mortality as a secondary endpoint.

There were some differences between the populations. Indonesian patients presented with more severe diseases (91.9% grade with grade II or III) than Vietnamese patients (62.2%). Also, CSF total protein, a proxy for blood–CSF barrier leakage (*Svensson et al., 2020*), was higher in Indonesian (median = 1.6 g/l, IQR = 0.8–3.1) than Vietnamese (1.3 g/l, IQR = 0.8–2.0) patients. CSF polymorphonuclear cell counts were higher in the Indonesian than in the Vietnamese patients where it showed a bimodal distribution associated to study site (*Figure 2—figure supplement 1*) and stratified analyses were performed taking this into account. Compared to the TBM patients, non-infectious controls (*n* = 54), bacterial meningitis patients (*n* = 50), and cryptococcal meningitis patients (*n* = 60) had a similar age range and gender distribution.

Ten metabolites showed detectable levels in >75% of patients and passed quality control, while two metabolites, 3-methoxyanthranilate and 5-hydroxyindoleacetic acid, were detected in less than 75% of patients and excluded from further analysis. Metabolite measurements showed stable concentrations over and were not affected by year of patient inclusion and duration of sample (*Figure 2—figure supplement 2*). The clinical metadata and LC–MS data before pre-processing can be found in *Source data 1*.

### Increased CSF tryptophan levels were associated with mortality of TBM patients independent of HIV status

Confirming our previous findings (*van Laarhoven et al., 2018*), higher baseline CSF tryptophan levels predicted 60-day survival in patients with TBM (hazard ratio [HR] = 1.16 for each doubling in CSF tryptophan, 95% confidence interval [CI] = 1.10–1.24), all analyses corrected for age, sex, and HIV status, and stratified for cohort (*Figure 2* and *Table 2*). This was both true for HIV-negative (HR = 1.13, 95% CI = 1.05–1.21) and HIV-positive patients (HR = 1.19, 95% CI = 1.07–1.33), who showed a much higher mortality (*Figure 2—figure supplement 3*), as reported previously (*Thuong et al., 2017*; *van*

**Table 1.** Patient baseline characteristics.

| | Tuberculous meningitis | Non-infectious control | Bacterial meningitis* | Cryptococcal meningitis* |
|---|---|---|---|---|
| | (n = 1069) | (n = 54) | (n = 50) | (n = 60) |
| **Clinical features** | | | | |
| Age, years | 34 (27–44) | 35 (25–44) | 46 (34–57) | 33 (27–37) |
| Sex, % male | 700 (65.5%) | 30 (55.6%) | 12 (60.0%) | 26 (78.8%) |
| Glasgow Coma Scale | 14 (12–15) | 15 (12–15) | 13 (9–14) | 15 (13–15) |
| HIV, % positive | 284 (26.6%) | 11 (20.4%) | 0 (0%) | 60 (100%) |
| **Tuberculous meningitis grade (%)** | | | | |
| Grade I | 287 (27.3%) | - | - | - |
| Grade II | 584 (55.6%) | - | - | - |
| Grade III | 179 (17.0%) | - | - | - |
| **Cerebrospinal fluid features** | | | | |
| Leukocytes, cells/µl | 150 (49–336) | 2 (1–3) | 1900 (739–5460) | 86 (24–192) |
| Neutrophils, cells/µl | 22 (3–99) | 1 (0–1) | 1527 (538–4986) | 17 (6–109) |
| Mononuclear cells, cells/µl | 98 (38–207) | 2 (1–3) | 307 (134–646) | 31 (6–89) |
| Protein, g/l | 1.46 (0.90–2.40) | 0.40 (0.26–0.59) | 1.90 (1.10–3.80) | 0.76 (0.58–1.60) |
| CSF to blood glucose ratio | 0.28 (0.17–0.40) | 0.60 (0.56–0.70) | 0.46 (0.17–1.00) | 0.50 (0.30–1.00) |
| *M. tuberculosis* culture or ZN staining or GeneXpert positive | 686 (64.17%) | - | - | - |
| **Outcomes** | | | | |
| **Outcome at day 60** | | | | |
| Alive | 825 (77.2%) | - | - | - |
| Deceased | 231 (21.6%) | - | - | - |
| Lost to follow-up | 13 (1.2%) | - | - | - |
| **Outcome at day 180** | | | | |
| Alive | 731 (68.4%) | - | - | - |
| Deceased | 304 (28.4%) | - | - | - |
| Lost to follow-up | 34 (3.2%) | - | - | - |

Categorical variables are presented in N (%); continuous variables are presented in median (IQR). Abbreviation: CSF = cerebrospinal fluid.

*Clinical metadata available for 40% of bacterial meningitis and 56% of cryptococcal meningitis patients.

*Laarhoven et al., 2017*). Because of weak negative correlation between tryptophan and GCS ($r = -0.08$) and to exclude the possibility that tryptophan is a marker of patients with more severe disease, we performed a post hoc analysis with two additional models. These models included the basic pre-defined variables (age, sex, HIV as covariates, study site as stratum). Adding GCS did not substantially change the effect size of CSF tryptophan (HR 1.15, 95% CI 1.08–1.23), and neither did adding GCS and CSF cell counts, CSF protein and glucose ratio (HR for CSF tryptophan 1.14, 95% CI 1.07–1.22) confirming that tryptophan was associated with mortality independent of these parameters. Baseline CSF tryptophan was associated with both early (HR = 1.14, 95% CI = 1.06–1.23) and late (HR = 1.17, 95% CI = 1.08–1.26) mortality (*Table 3*). Compared to non-infectious controls, CSF tryptophan was lower. This was also observed in patients with cryptococcal, but not in those with bacterial meningitis (*Figure 3*). Interestingly, among 17 cryptococcal meningitis patients with available in-hospital

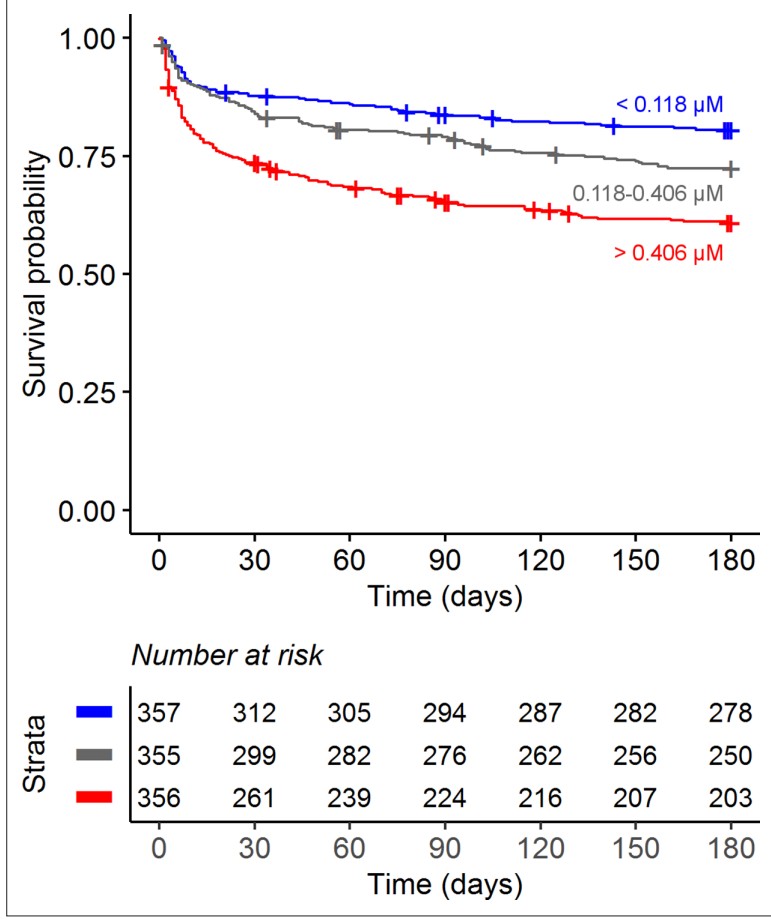

**Figure 2.** Six-month survival curve of tuberculous meningitis (TBM) patients. Patients were stratified by cerebrospinal fluid (CSF) tryptophan concentrations tertiles.

The online version of this article includes the following figure supplement(s) for figure 2:

**Figure supplement 1.** Cerebrospinal fluid (CSF) parameters of tuberculous meningitis (TBM) patients in Indonesia and Vietnam.

**Figure supplement 2.** Stability of metabolites over-time.

**Figure supplement 3.** Six-month survival curve of tuberculous meningitis (TBM) patients stratified by HIV status.

**Figure supplement 4.** In-hospital mortality for 17 HIV-positive patients with cryptococcal meningitis.

**Figure supplement 5.** Cerebrospinal fluid (CSF) tryptophan distributions according to mycobacterial load.

mortality data in Indonesia, baseline CSF tryptophan was significantly higher in those who died in hospital compared to those discharged alive (*Figure 2—figure supplement 4*), similar as in TBM.

## CSF tryptophan levels do not reflect mycobacterial burden

We next examined if CSF tryptophan was associated with CSF mycobacterial load. We hypothesized that a low baseline tryptophan might either reflect a lower bacterial load, as *M. tuberculosis* can produce tryptophan, *or* might cause a lower bacterial load as tryptophan depletion impairs mycobacterial growth (*Zhang et al., 2013*). Instead, we found a reverse, albeit weak relationship: tryptophan was higher in CSF culture-negative (median = 0.31 µM) than culture-positive (median = 0.14 µM, p < 0.001) TBM patients. Similarly, among patients with CSF GeneXpert-confirmed TBM patients, we did not find a correlation between CSF tryptophan and quantitative PCR results (Spearman's rho = 0.084, p = 0.105, *Figure 2—figure supplement 5*). Interestingly, within patients with microbiologically confirmed TBM the effect of tryptophan was stronger (HR = 1.28, 95% CI = 1.17–1.40) than in patients with probable TBM (HR = 1.07, 95% CI = 0.98–1.18).

**Table 2.** Univariate Cox regression for influence of cerebrospinal fluid (CSF) metabolites on 60-day mortality.

| Metabolites | HR* | 95% CI* | p-value | FDR[†] |
|---|---|---|---|---|
| Tryptophan | 1.16 | 1.10, 1.24 | <0.001 | **<0.001** |
| Kynurenine | 1.00 | 0.93, 1.07 | >0.9 | >0.9 |
| Kynurenic acid | 1.00 | 0.93, 1.07 | 0.9 | >0.9 |
| 3-Hydroxyanthranilic acid | 1.01 | 0.97, 1.05 | 0.6 | 0.9 |
| Xanthurenic acid | 0.95 | 0.90, 1.00 | 0.05 | 0.2 |
| Quinolinic acid | 0.92 | 0.85, 1.00 | 0.038 | 0.2 |
| Niacinamide | 1.03 | 0.95, 1.11 | 0.5 | 0.8 |
| 3-Indoleacetic acid | 1.11 | 0.96, 1.29 | 0.2 | 0.4 |
| N-Acetylserotonin | 1.01 | 0.94, 1.09 | 0.7 | 0.9 |
| 5-Methoxytryptophol | 1.11 | 0.93, 1.32 | 0.3 | 0.5 |

Baseline cerebrospinal fluid (CSF) metabolites were measured in 1069 TBM patients. Cox regression models were stratified by cohort and adjusted by sex, age, and HIV status. Hazard ratio (HR) was calculated per twofold increase in metabolite concentration. Bold: False Discovery Rate (FDR) < 0.05.
*HR = hazard ratio, CI = confidence interval.
[†]FDR = false discovery rate; Benjamini and Hochberg correction for multiple testing.

## Relationship between cerebral and systemic metabolism and its impact on survival

Ninety-five percent of tryptophan is converted to kynurenine (*Lovelace et al., 2017*) and we therefore hypothesized that lower CSF tryptophan levels in TBM are caused by higher conversion to kynurenine, and that the higher CSF tryptophan associated with death could reflect reduced activity of IDO1 and other downstream enzymes. CSF kynurenine (*Figure 3*) and its downstream metabolite kynurenic acid (*Figure 3—figure supplement 1*) were higher in TBM patients, bacterial meningitis and cryptococcal

**Table 3.** Univariate Cox regression for influencen of baseline cerebrospinal fluid (CSF) metabolites on early and late mortality.

| Metabolite | Early mortality (days 0–14) | | | | Late mortality (days 14–180) | | | |
|---|---|---|---|---|---|---|---|---|
| | HR* | 95% CI* | p-value | FDR[†] | HR* | 95% CI* | p-value | FDR[†] |
| Tryptophan | 1.14 | 1.06, 1.23 | <0.001 | **0.005** | 1.17 | 1.08, 1.26 | <0.001 | **<0.001** |
| Kynurenine | 1.03 | 0.95, 1.13 | 0.4 | 0.6 | 1 | 0.91, 1.10 | >0.9 | >0.9 |
| Kynurenic acid | 1.05 | 0.96, 1.14 | 0.3 | 0.5 | 0.95 | 0.86, 1.04 | 0.3 | 0.5 |
| 3-Hydroxyanthranilic acid | 1.02 | 0.97, 1.06 | 0.5 | 0.6 | 1.01 | 0.96, 1.06 | 0.6 | 0.7 |
| Xanthurenic acid | 0.96 | 0.90, 1.03 | 0.2 | 0.5 | 0.96 | 0.89, 1.04 | 0.3 | 0.5 |
| Quinolinic acid | 0.89 | 0.81, 0.98 | 0.02 | 0.1 | 0.9 | 0.81, 1.00 | 0.052 | 0.2 |
| Niacinamide | 1.02 | 0.92, 1.12 | 0.7 | 0.7 | 1.05 | 0.95, 1.16 | 0.3 | 0.5 |
| 3-Indoleacetic acid | 1.1 | 0.92, 1.32 | 0.3 | 0.5 | 1.18 | 0.97, 1.43 | 0.093 | 0.2 |
| N-Acetylserotonin | 1.05 | 0.96, 1.14 | 0.3 | 0.5 | 0.96 | 0.88, 1.06 | 0.4 | 0.5 |
| 5-Methoxytryptophol | 1.09 | 0.87, 1.35 | 0.5 | 0.6 | 1.29 | 1.04, 1.59 | 0.02 | 0.1 |

Baseline cerebrospinal fluid (CSF) metabolites were measured in 1069 TBM patients. Cox regression models were stratified by sites and adjusted by age, sex, and GCS. Bold: False Discovery Rate (FDR) < 0.05.
*HR = hazard ratio, CI = confidence interval.
[†]Benjamini and Hochberg correction for multiple testing.

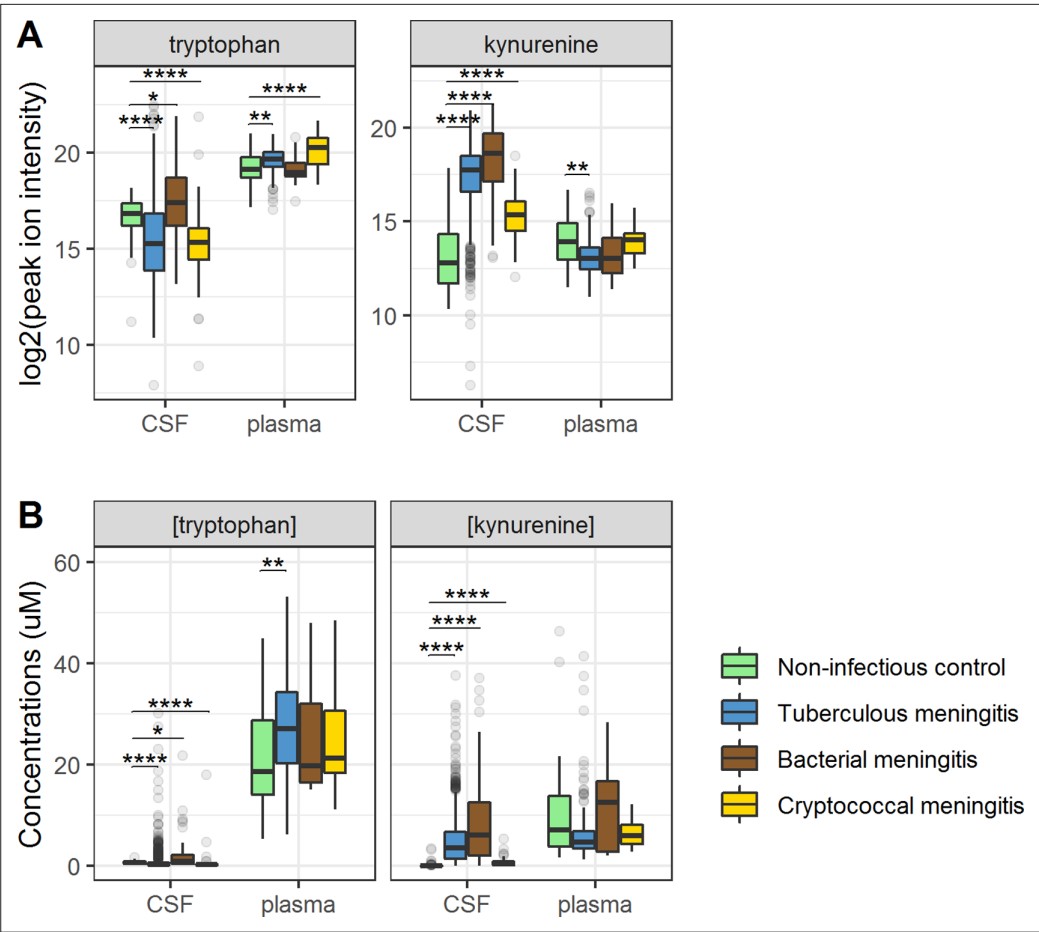

**Figure 3.** Cerebrospinal fluid and plasma metabolites concentrations in tuberculous meningitis (TBM) and all controls for the tryptophan metabolites associated with outcome: tryptophan and kynurenine. (**A**) Relative concentrations based on peak ion intensity and (**B**) absolute concentrations in µM. Boxplots are shown with Holm-adjusted Wilcoxon rank-sum test p-values are presented: (*p<0.05, **p<0.01, ***p<0.001, ****p<0.0001). No measurements were excluded for these graphs. Note: plasma measurements were available for a subset of 300 TBM patients and all controls.

The online version of this article includes the following figure supplement(s) for figure 3:

**Figure supplement 1.** Boxplots of cerebrospinal fluid (CSF) and plasma metabolites concentrations in tuberculous meningitis (TBM) and controls.

meningitis patients compared to non-infectious controls, but not significantly different between surviving and non-surviving TBM patients (*Table 2*).

Then, to examine the relation between cerebral and systemic tryptophan metabolism, we compared concentrations of CSF metabolites with those in plasma, measured in a subset of 300 TBM patients. In contrast to our findings in CSF, plasma tryptophan levels were higher and kynurenine levels were lower in TBM patients compared to controls. As the CSF kynurenine metabolites positively correlated with CSF protein (*Figure 4*), a proxy for barrier leakage (*Svensson et al., 2020*), we hypothesized that systemic leakage might be an additional source of kynurenine. For a subset of metabolites, absolute quantification of metabolite levels was achieved. This showed that the increase in CSF kynurenine in TBM patients ($\Delta$ = 3.52 µM) was much more marked than the decrease in CSF tryptophan ($\Delta$ = 0.39 µM, *Figure 3B*). Corroborating our leakage hypothesis, the CSF–plasma gradient of the kynurenine metabolites correlated positively with total CSF protein (*Figure 4—figure supplement 1*). Plasma tryptophan did not predict mortality, but plasma levels of its downstream metabolites kynurenine strongly predicted mortality (*Table 4*).

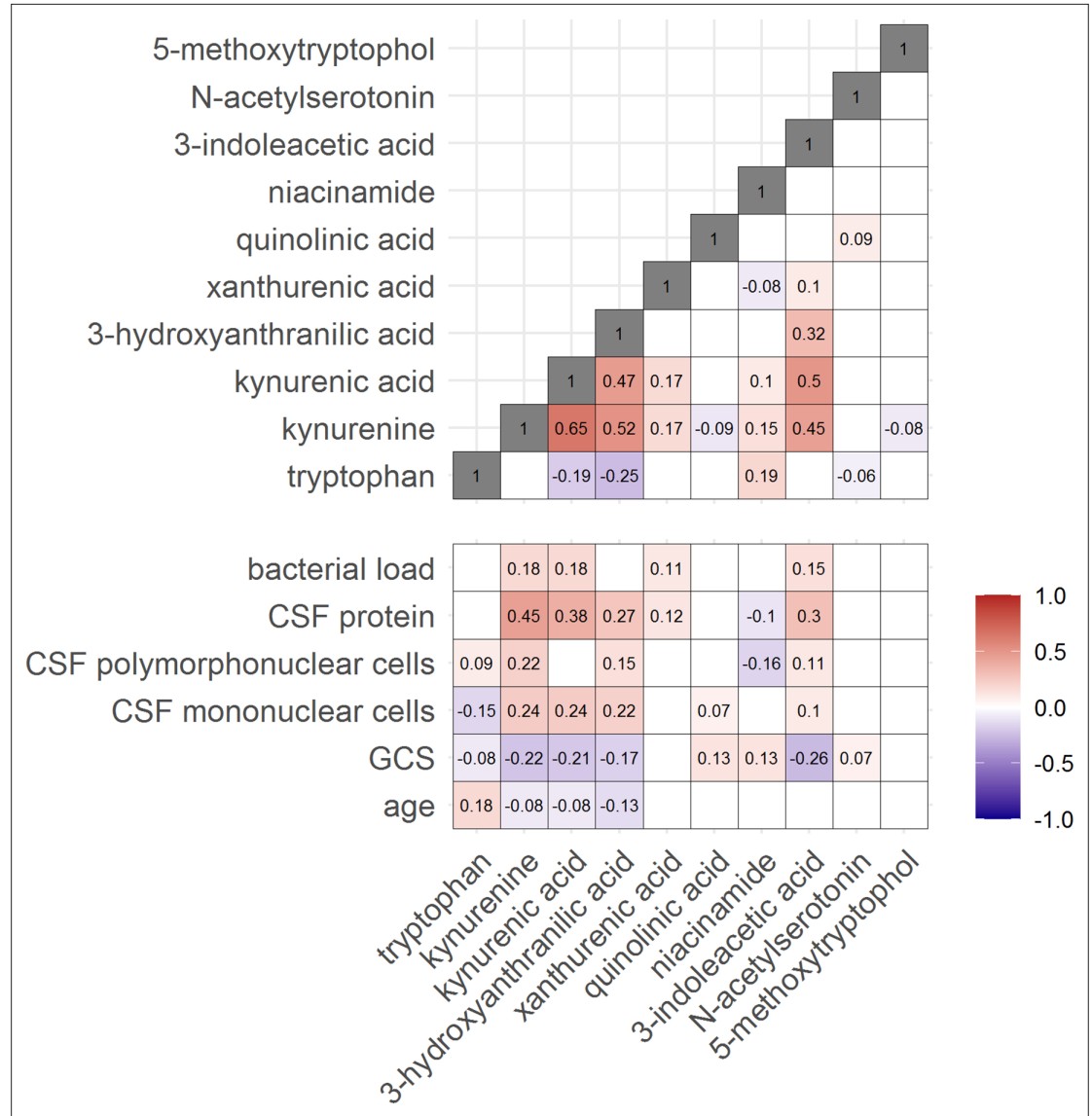

**Figure 4.** Correlation between tryptophan metabolites and with clinical and cerebrospinal fluid (CSF) parameters. Significant Spearman's correlation coefficients are presented in the correlation matrix, while the ones with not significant correlations were blank. Red indicates positive correlations, and blue indicates negative ones. The color gradient shows the strength of the associations.

The online version of this article includes the following figure supplement(s) for figure 4:

**Figure supplement 1.** Associations between cerebrospinal fluid (CSF)/plasma metabolite ratios (*y*-axis) and CSF protein levels (as a proxy of CSF barrier leakage, *x*-axis).

## CSF tryptophan is inversely correlated with interferon gamma

We next looked at correlations of tryptophan metabolites and inflammation, as inflammation is a determinant of outcome from TBM (*Wilkinson et al., 2017*). Tryptophan is transported into the brain by the large neutral amino acid transporters (*Boado et al., 1999*). Out of 92 inflammatory proteins measured in CSF from 176 TBM patients from Indonesia, 80 proteins were detectable in >75% of patients. Tryptophan correlated inversely to a small cluster of 13 cytokines, including interferon gamma (IFN-γ, *r* = −0.48, *Figure 5—figure supplement 1A*). In line with this finding, the 10 measured CSF cytokines in the Vietnamese patients showed a very similar pattern (*Figure 5—figure supplement 1B*) and a higher CSF IFN-γ has previously been shown to predict survival of Vietnamese TBM patients (*Thuong et al., 2017*). IFN-γ is known to induce IDO1 (*Zhang et al., 2013*), which converts tryptophan to kynurenine. We indeed confirmed the inverse correlation between CSF tryptophan and IFN-γ in our Vietnamese

**Table 4.** Univariate Cox regression for influence of baseline plasma metabolites on 60-day mortality.

| Metabolite | HR* | 95% CI* | p-value | FDR† |
|---|---|---|---|---|
| Tryptophan | 0.8 | 0.56, 1.16 | 0.2 | 0.4 |
| Kynurenine | 1.54 | 1.22, 1.93 | <0.001 | **0.002** |
| Kynurenic acid | 1.2 | 1.01, 1.43 | 0.036 | 0.2 |
| 3-Hydroxyanthranilic acid | 1.13 | 1.00, 1.28 | 0.045 | 0.2 |
| Xanthurenic acid | 1.11 | 0.98, 1.25 | 0.1 | 0.3 |
| Quinolinic acid | 0.99 | 0.88, 1.12 | 0.9 | 0.9 |
| Niacinamide | 0.92 | 0.79, 1.07 | 0.3 | 0.4 |
| 3-Indoleacetic acid | 1.04 | 0.97, 1.12 | 0.3 | 0.4 |
| *N*-Acetylserotonin | 1.05 | 0.91, 1.22 | 0.5 | 0.6 |
| 5-Methoxytryptophol | 1.04 | 0.85, 1.28 | 0.7 | 0.8 |

Baseline plasma tryptophan metabolites were measured in a subset 300 patients. Cox regression models were stratified by sites and adjusted by age, sex, and GCS. Bold: False Discovery Rate (FDR) < 0.05.
*HR = hazard ratio, CI = confidence interval.
†Benjamini and Hochberg correction for multiple testing.

patients (Spearman's rho = −0.45, p < 0.0001, *Figure 5A*), irrespective of HIV status. Different from tryptophan, the kynurenine metabolites (kynurenine, kynurenic acid, 3-hydroxyanthranilic acid, and quinolinic acid) correlated positively with a large cluster of inflammatory proteins including the hallmark inflammatory protein tumor necrosis factor alpha (TNF-α; *r* = 0.30 for kynurenine), which we could again confirm in the Vietnamese patients (Spearman's rho = 0.30, p < 0.0001, *Figure 5B*).

## Discussion

We previously found that CSF concentrations of tryptophan were lower in HIV-negative Indonesian adults with TBM compared to non-infectious controls, and that TBM patients with lower tryptophan levels had lower mortality (*van Laarhoven et al., 2018*). In the current study, we confirm these observations in a much larger cohort of HIV-negative and -positive patients from both Vietnam and Indonesia. Aiming to understand how tryptophan metabolism is altered in TBM and how it might exert its effect on patient outcome, we correlated its concentrations with bacterial load and CSF inflammatory markers and measured downstream metabolites in both CSF and plasma. Our findings show that CSF concentrations of downstream kynurenine metabolites did not predict mortality, and that higher tryptophan levels were not associated with a higher bacterial load. Also, while kynurenine metabolites strongly correlated with CSF inflammatory markers and CSF protein, a marker of blood–CSF leakage, there was no association with CSF tryptophan. Tryptophan did however show a negative correlation with IFN-γ, important for immunity against mycobacteria. Collectively, these findings suggest that a process involving tryptophan metabolism affects outcome from TBM. Given the lack of plasma tryptophan with outcome, this process may take place in the brain rather than systemically. It is potentially driven by IFN-γ but not associated with nonspecific inflammation, and independent from downstream tryptophan metabolism or bacterial replication. In contrast, kynurenine may affect outcome systemically by leakage across the blood–brain barrier.

CSF tryptophan increases with age in individuals without central nervous system infections (*Hestad et al., 2017*). Age is known to negatively impact outcome of TBM and in this study, higher age was associated with higher CSF tryptophan concentrations. All mortality analyses were therefore corrected for age, as well as sex and HIV status, and analysis was stratified for country because of the overall higher mortality in Indonesian compared to Vietnamese TBM patients (*Thuong et al., 2017*; *van Laarhoven et al., 2017*). We further tested whether higher CSF tryptophan reflected a higher mycobacterial burden and refuted this hypothesis. For cryptococcal meningitis, no previous data on cerebral tryptophan metabolism were known. These patients follow a pattern similar to TBM, with low

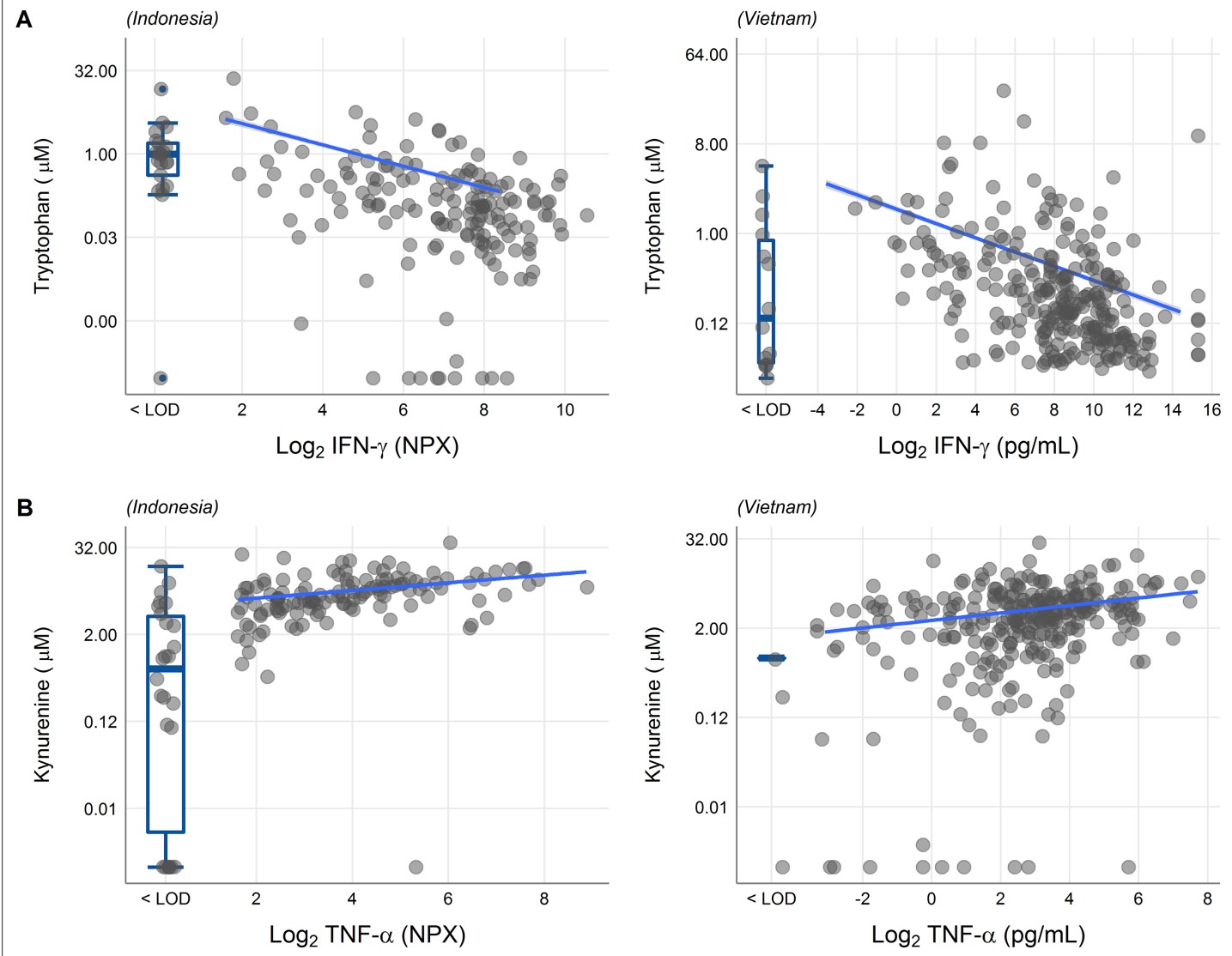

**Figure 5.** Associations of cerebrospinal fluid (CSF) tryptophan with IFN-γ (**A**) and with TNF-α (**B**) in 176 Indonesian (left) and 304 (Vietnamese) tuberculous meningitis (TBM) patients. The boxplots on the left of each plot show the association of metabolites with cytokines below the detection limit. Abbreviations: IFN-γ: interferon gamma, TNF-α: tumor necrosis factor alpha, LOD: lower limit of detection.

The online version of this article includes the following figure supplement(s) for figure 5:

**Figure supplement 1.** Correlation between cerebrospinal fluid (CSF) tryptophan metabolites and inflammatory markers measured with O-link in Indonesian (**A**) and Vietnamese (**B**) tuberculous meningitis patients.

tryptophan and high kynurenine, and in a small number of cryptococcal meningitis patients, a high baseline CSF tryptophan predicted mortality, similar as for TBM.

Systemic tryptophan and kynurenine are transported into the brain over the large amino acid transporter *LAT1* (**Boado et al., 1999**). In a healthy brain, systemic and CSF kynurenine positively correlate, as do CSF concentrations of tryptophan and kynurenine (**Hestad et al., 2017**). In patients with cerebral inflammation, the correlation between CSF kynurenine and tryptophan can be lost, probably through increased catabolism through IDO upregulation, which also has been demonstrated in the brain parenchyma of deceased TBM patients (**Kumar et al., 2012**). Although we found low CSF tryptophan and high CSF kynurenine in TBM compared to healthy controls, the two did not intercorrelate and moreover, the increase in CSF kynurenine was much larger than the decrease in CSF tryptophan and it is therefore unlikely that upregulation of IDO1 solely explains this which precludes catabolism as the sole explanation. This suggests that increased blood to central nervous system kynurenine

transport as an additional mechanism to IDO1 upregulation. Endothelial cells and pericytes of the blood–brain barrier can upregulate tryptophan catabolism into kynurenine metabolites upon IFN-γ stimulation (*Owe-Young et al., 2008*). Our findings corroborate this hypothesis because we find a negative correlation between CSF IFN-γ and CSF tryptophan in our patients.

We examined whether higher CSF tryptophan concentrations reflected higher concentrations of downstream kynurenine metabolites that may have neurotoxic (quinolinic acid) or lower levels of the metabolites that may have neuroprotective (kynurenic acid) properties (*Lovelace et al., 2017*) and refuted these hypotheses. Interestingly however, CSF kynurenine metabolites correlated with CSF cell counts and pro-inflammatory proteins, including TNF-α. Kynurenine is sensed by the aryl hydrocarbon receptor (AhR), which is important for the upregulation of TNF among other pro-inflammatory cytokines in a mouse model (*Moura-Alves et al., 2014*), in line with our CSF findings. The increased CSF kynurenine levels we found in TBM have been reported before in bacterial meningitis (*Coutinho et al., 2014*; *Sühs et al., 2019*) and in cerebral malaria (*Medana et al., 2003*) and in plasma from pulmonary TB patients (*Weiner et al., 2018*). Of interest, nicotinamide can inhibit *M. tuberculosis* growth, and can compete with isoniazid for antimycobacterial effects (*Murray, 2003*). We did however not find a detrimental effect of a higher nicotinamide, possibly because of its complex biology, that is it can also be produced by *M. tuberculosis* when human dietary niacin intake is limited (*Adu-Gyamfi et al., 2019*).

Strengths of our study include the large numbers of clinically well-phenotyped patients from multiple independent study sites in Indonesia and Vietnam, including a significant proportion of HIV-positive patients. We moreover used a sensitive triple quadrupole (QQQ) mass spectrometry method specifically designed to accurately target the tryptophan metabolites. Absolute quantification of a subset of metabolites further facilitated interpretation. Due to differences in polarity of the downstream tryptophan metabolites, we could not measure the complete tryptophan pathway. The availability of CSF at baseline only, limits our ability to understand how changes in tryptophan metabolism influence mortality. And we infer our observations from lumbar CSF, which reflects biological processes from both the blood and the brain. The use of ventricular CSF could help establishing what processes in the brain parenchyma take place. Moreover, although the lower CSF tryptophan values in TBM patients compared to non-infectious controls, implies active tryptophan metabolism in TBM, definite claims can only be made in an interventional study. Therefore animal studies, preferably combined with live imaging (*Mota et al., 2022*), are warranted to see whether pharmacological induction of IDO1 (for instance with recombinant IFN-γ), or inhibitory tryptophan analogues (*Wang et al., 2021*) should be priorities as adjuvant therapeutic candidates for future personalized trials.

In summary, we confirm the importance of CSF tryptophan to outcome from HIV-negative and -positive adults with TBM, independent from downstream kynurenine metabolism, bacterial load, and inflammation. We additionally show the potential importance of systemic kynurenine as a predictor of mortality. Better understanding of the metabolic pathways associated with TBM may lead to more targeted therapies, as adjuvant immunotherapy may modulate the aberrant metabolic pathways and thus improve outcome.

## Materials availability

The clinical metadata and LC–MS data before pre-processing are available in *Source data 1*.

## Acknowledgements

The authors thank the neurology residents and Tiara Pramaesya, Sofia Immaculata, Putri Andini, Sri Margi, Rani Trisnawati, and Shehika Shulda of the tuberculous meningitis study team for monitoring patients and data management; Lidya Chaidir and Jessi Annisa for mycobacterial diagnostics; the director of the Hasan Sadikin General Hospital, Bandung, Indonesia, for accommodating the research. We also express our gratitude to our funders: This study was supported by National Institutes of Health (R01AI145781) and the Wellcome Trust (110179/Z/15/Z and 206724/Z/17/Z). Previous establishment of the cohorts in Indonesia was supported by the Direktorat Jendral Pendidikan Tinggi (BPPLN fellowship to SD) and the Ministry of Research, Technology, and Higher Education, Indonesia (PKSLN grant to RR and SD), and United States Agency for International Development (PEER Health grant to RR). The funders had no role in study design, data collection and analysis, decision to publish, or preparation of the manuscript.

# Additional information

## Competing interests

Mihai G Netea: has received consulting fees from Scientific Board TTxD and is a scientific founder of TTxD, Lemba and BioTRIP. The author has no other competing interests to declare. The other authors declare that no competing interests exist.

## Funding

| Funder | Grant reference number | Author |
| --- | --- | --- |
| National Institutes of Health | R01AI145781 | Edwin Ardiansyah<br>Julian Avila-Pacheco<br>Le Thanh Hoang Nhat<br>Sofiati Dian<br>Dao Nguyen Vinh<br>Hoang Thanh Hai<br>Kevin Bullock<br>Bachti Alisjahbana<br>Mihai G Netea<br>Riwanti Estiasari<br>Trinh Thi Bich Tram<br>Joseph Donovan<br>Dorothee Heemskerk<br>Tran Thi Hong Chau<br>Nguyen Duc Bang<br>Ahmad Rizal Ganiem<br>Valerie ACM Koeken<br>Raph L Hamers<br>Darma Imran<br>Kartika Maharani<br>Vinod Kumar<br>Clary B Clish<br>Reinout van Crevel<br>Guy Thwaites<br>Arjan van Laarhoven<br>Nguyen Thuy Thuong Thuong |
| Wellcome Trust | Africa and Asia Programme | Le Thanh Hoang Nhat<br>Dao Nguyen Vinh<br>Hoang Thanh Hai<br>Trinh Thi Bich Tram<br>Joseph Donovan<br>Dorothee Heemskerk<br>Tran Thi Hong Chau<br>Nguyen Duc Bang<br>Guy Thwaites<br>Nguyen Thuy Thuong Thuong |
| Wellcome Trust | Intermediate Fellowship 110179/Z/15/Z | Nguyen Thuy Thuong Thuong |
| Direktorat Jenderal Pendidikan Tinggi | BPPLN | Sofiati Dian |
| Ministry of Research, Technology and Higher Education of the Republic of Indonesia | PKSLN | Sofiati Dian<br>Rovina Ruslami |
| United States Agency for International Development | PEER Health | Rovina Ruslami |

The funders had no role in study design, data collection, and interpretation, or the decision to submit the work for publication.

## Author contributions

Edwin Ardiansyah, Conceptualization, Data curation, Formal analysis, Visualization, Methodology, Writing - original draft, Writing – review and editing; Julian Avila-Pacheco, Conceptualization, Data

curation, Supervision, Methodology, Writing – review and editing; Le Thanh Hoang Nhat, Formal analysis, Supervision, Writing – review and editing; Sofiati Dian, Supervision, Investigation, Writing – review and editing; Dao Nguyen Vinh, Hoang Thanh Hai, Formal analysis; Kevin Bullock, Riwanti Estiasari, Trinh Thi Bich Tram, Dorothee Heemskerk, Tran Thi Hong Chau, Nguyen Duc Bang, Investigation; Bachti Alisjahbana, Conceptualization, Supervision; Mihai G Netea, Conceptualization, Supervision, Funding acquisition; Joseph Donovan, Investigation, Writing – review and editing; Ahmad Rizal Ganiem, Rovina Ruslami, Valerie ACM Koeken, Raph L Hamers, Supervision; Darma Imran, Kartika Maharani, Supervision, Investigation; Vinod Kumar, Nguyen Thuy Thuong Thuong, Conceptualization, Supervision, Writing – review and editing; Clary B Clish, Reinout van Crevel, Guy Thwaites, Conceptualization, Supervision, Funding acquisition, Writing – review and editing; Arjan van Laarhoven, Conceptualization, Data curation, Supervision, Visualization, Writing – review and editing

## Author ORCIDs

Edwin Ardiansyah ⓘ http://orcid.org/0000-0003-2702-5497
Le Thanh Hoang Nhat ⓘ http://orcid.org/0000-0002-3756-7217
Sofiati Dian ⓘ http://orcid.org/0000-0001-9909-4171
Hoang Thanh Hai ⓘ http://orcid.org/0000-0002-1809-5541
Mihai G Netea ⓘ http://orcid.org/0000-0003-2421-6052
Valerie ACM Koeken ⓘ http://orcid.org/0000-0002-5783-9013
Guy Thwaites ⓘ http://orcid.org/0000-0002-2858-2087
Arjan van Laarhoven ⓘ http://orcid.org/0000-0002-6607-4075
Nguyen Thuy Thuong Thuong ⓘ http://orcid.org/0000-0001-8733-692X

## Ethics

Ethical approval was obtained from the Ethical Committee of Hasan Sadikin Hospital, Faculty of Medicine, Universitas Padjadjaran, Bandung, Indonesia and from the Oxford Tropical Research Ethics Committee in the United Kingdom, the Institutional Review Boards of the Hospital for Tropical Diseases and Pham Ngoc Thach Hospital in Vietnam. Written (Vietnam) or oral (Indonesia) consent to be included in the study, for storage of surplus sample, and to obtain follow-up data was obtained from patients or close relatives of patients who were unconscious.

## Decision letter and Author response

Decision letter https://doi.org/10.7554/eLife.85307.sa1
Author response https://doi.org/10.7554/eLife.85307.sa2

---

# Additional files

## Supplementary files

- MDAR checklist
- Source data 1. The source data includes clinical metadata and LC-MS data before pre-processing.
- Reporting standard 1.

## Data availability

The data generated or analyzed during this study are included in the supporting file.

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
