## [Editor Report]

This important study by Ardiansyah and colleagues reports the association of tryptophan levels in cerebrospinal fluid with 60-day mortality in patients with tuberculosis meningitis. Good evidence is presented that cerebrospinal fluid tryptophan levels are associated with mortality. The findings continue to remain an association, without clarity of whether tryptophan is a key mediator of mortality or another inflammatory biomarker. The work will be of interest to tuberculosis researchers.

---

## [Decision Letter]

**Decision letter after peer review:**

Thank you for submitting your article "Tryptophan metabolism determines outcome in tuberculous meningitis: a targeted metabolomic analysis" for consideration by *eLife*. Your article has been reviewed by 2 peer reviewers, and the evaluation has been overseen by a Reviewing Editor and Bavesh Kana as the Senior Editor. The following individual involved in review of your submission has agreed to reveal their identity: Diederik van de Beek (Reviewer #1).

Essential revisions:

1. Modelling. The impact of baseline CSF and plasma metabolite levels on 60-day survival was tested in a Cox-regression model, adjusted for sex, age, and HIV. One of the research questions could be whether tryptophan concentrations and outcome in TBM is a defining factor for outcome or just simply a bystander effect of damage. To study this in a cohort with >1000 TBM patient and 231 deaths, the authors might want to perform multiple regression analyses for 30-day survival, including baseline and clinical features presented in the baseline table. They could include tryptophan concentrations in categories depending on the distribution of values (i.e., quartiles) and perform multiple imputation for missing values. While the association of CSF tryptophan levels with mortality (high tryptophan levels associated with higher 60-day mortality) are an interesting finding, it is confusing to note that uninfected control subjects had much higher CSF tryptophan levels (Figure 3). Similar, trend to TB meningitis (i.e. higher mortality in those with higher tryptophan levels) were noted in patients with cryptococcal meningitis. Finally, patients with bacterial meningitis also had much higher tryptophan levels but information on mortality is lacking. Therefore, it is likely that CSF tryptophan levels may not be the driver of mortality and more likely is an inflammatory biomarker in chronic forms of meningitis. This needs more detailed analysis and discussion.

2. Study power. Based on the previous findings and the number of patients in the cohorts, the authors could say something about the statistical power, i.e., what are changes that they are able to detect.

3. Early vs. late mortality. The association between tryptophan concentrations and outcome in TBM seems to be not only true for the early period (e.g., the first 14 days), but also for the late period. As suggested in the results (HIV+ vs HIV- patients; suppl table 1). Could the authors explain in more detail how these analyses were performed? It is difficult to understand how base-line metabolites have a strong effect on late outcome (i.e., at day 180). This might even point towards the hypothesis that found metabolites are not the causing factor of death but more just a bystander effect.

4. CSF tryptophan is inversely correlated with interferon-γ. 92 inflammatory proteins measured in CSF from 176 TBM patients. 13 were inversely associated with tryptophan concentrations, only interferon-γ was validated in Vietnamese samples. What about the other inflammatory proteins?

5. Defining factor in outcome or bystander. In the first paragraph of the discussion the authors overstate the impact of their results. This is a pity because it is not needed. They state that "Collectively, these finding suggest that tryptophan affects outcome from TBM within the brain rather than systemically. Results do not support that tryptophan affects outcome; they support the statement that tryptophan is associated with outcome.

6. Negative correlation between tryptophan and IFN. The authors state that they find a strong negative correlation between tryptophan and IFN. The correlation found between tryptophan and IFN (which indeed is stronger than the correlation in Indonesia) has a Spearman's rho of 0.30. Correlation coefficients between 0.3 and 0.5 indicate variables which have a low correlation, not a strong correlation. So, it would be more correct to state that "a higher tryptophan showed a statistically significant but low correlation with IFN-γ". Please correct throughout results and discussion. Better not to overstate your results.

7. There is a large variability in CSF tryptophan levels. If a cutoff were defined to predict the mortality risk, what would be PPV and NPV of this test?

8. There was some discordance between the plasma and CSF levels of tryptophan and its downstream metabolites. What is known about the passage of tryptophan and its metabolites across the blood-brain barrier and inflammation? What would a multi-variate analysis utilizing markers of CSF inflammation (or IFN-g as noted by the authors) show? Would tryptophan CSF level still remain as an independent predictor of mortality?

9. It has been shown previously that ventricular and spinal CSF have different profiles (and acknowledged by the authors in the current manuscript). Since spinal CSF was utilized by the current studies, it would be interesting to know how many of the patients in this cohort also had spinal TB and if this was correlated with tryptophan/metabolite levels.

10. The authors suggest the use of animal models, but do not quote any for TB meningitis, but which could be quoted (PMID: 29777209, 32501258, 30518610, 35085105, 36581633).

*Reviewer #1 (Recommendations for the authors):*

Previously, lower cerebrospinal fluid tryptophan concentrations were associated with survival of TBM patients in Indonesia. This study confirms this previous finding. In addition, downstream metabolites were measured in cerebrospinal fluid and blood samples of TBM patients and controls included in multiple cohorts. These findings confirm the association between cerebrospinal fluid tryptophan concentrations and outcome in TBM.

The strength of this manuscript is the validation aspect in several cohorts. the analyses have been carefully performed. I do have some suggestions for additional analyses and would like to advise the authors to not overstate their findings.

The weakness is that it remains unclear whether the association between cerebrospinal fluid tryptophan concentrations and outcome in TBM is a defining factor in outcome or just simply a bystander effect of damage.

Suggestions for additional analyses:

One of the research questions could be whether tryptophan concentrations and outcome in TBM is a defining factor in outcome or just simply a bystander effect of damage. To study this in a cohort with >1000 TBM patient and 231 deaths, you might want to perform multiple regression analyses for 30-day survival, including baseline and clinical features presented in the baseline table.

Presentation

I am not so convinced about the "strong correlation" between tryptophan and IFN. The correlation found between tryptophan and IFN (which indeed is stronger than the correlation in Indonesia) has a Spearman's rho of 0.30 (with I consider as a weak correlation).

This is a good and solid confirmatory study. No need to overstate your findings. I have some suggestions and recommendations for improvement:

Modelling. The impact of baseline CSF and plasma metabolite levels on 60-day survival was tested in a Cox-regression model, adjusted for sex, age, and HIV. One of the research questions could be whether tryptophan concentrations and outcome in TBM is a defining factor in outcome or just simply a bystander effect of damage. To study this in a cohort with >1000 TBM patient and 231 deaths, you might want to perform multiple regression analyses for 30-day survival, including baseline and clinical features presented in the baseline table. You could include tryptophan concentrations in categories depending on the distribution of values (i.e., quartiles) and perform multiple imputation for missing values.

Study power. Based on your previous findings and the number of patients in the cohorts you could say something about the statistical power, i.e., what are changes that you are able to detect.

Early vs. late mortality. The association between tryptophan concentrations and outcome in TBM seems to be not only true for the early period (e.g., the first 14 days), but also for the late period. As suggested in the results (HIV+ vs HIV- patients; suppl table 1). Could you explain in more detail how these analyses were performed? It is difficult to understand how base-line metabolites have a strong effect on late outcome (i.e., at day 180). This might even point towards the hypothesis that found metabolites are not the causing factor of death but more just a bystander effect.

CSF tryptophan is inversely correlated with interferon-γ. 92 inflammatory proteins measured in CSF from 176 TBM patients. 13 were inversely associated with tryptophan concentrations, only interferon-γ was validated in Vietnamese samples. What about the other inflammatory proteins?

Defining factor in outcome or bystander. In the first paragraph of the discussion the authors overstate the impact of their results. This is a pity because it is not needed. They state that "Collectively, these finding suggest that tryptophan affects outcome from TBM within the brain rather than systemically. Results do not support that tryptophan affects outcome; they support the statement that tryptophan is associated with outcome. I suggest not to overstate findings.

Negative correlation between tryptophan and IFN. The authors state that they find a strong negative correlation between tryptophan and IFN. The correlation found between tryptophan and IFN (which indeed is stronger than the correlation in Indonesia) has a Spearman's rho of 0.30. Correlation coefficients between 0.3 and 0.5 indicate variables which have a low correlation, not a strong correlation. So, it would be more correct to state that "a higher tryptophan showed a statistically significant but low correlation with IFN-γ". Please correct throughout results and discussion. Better not to overstate your results.

*Reviewer #2 (Recommendations for the authors):*

This manuscript by Ardiansyah et al. reports the association of CSF tryptophan levels with 60-day mortality in patients with TB meningitis. This work is a follow up of a much smaller study in patients with TB meningitis in Indonesia, reporting the same findings and performed by some authors from this study. The current study is larger, includes patients with HIV co-infection as well as has control groups with individuals with other etiologies of meningitis. Suggestions for the manuscript are below.

1. While the association of CSF tryptophan levels with mortality (high tryptophan levels associated with higher 60-day mortality) are an interesting finding, it is confusing to note that uninfected control subjects had much higher CSF tryptophan levels (Figure 3). Similar, trend to TB meningitis (i.e. higher mortality in those with higher tryptophan levels) were noted in patients with cryptococcal meningitis. Finally, patients with bacterial meningitis also had much higher tryptophan levels but information on mortality is lacking. Therefore, it is likely that CSF tryptophan levels may not be the driver of mortality and more likely is an inflammatory biomarker in chronic forms of meningitis. This needs more detail in the discussion.

2. There is a large variability in CSF tryptophan levels. If a cutoff were defined to predict the mortality risk, what would be PPV and NPV of this test?

3. There was some discordance between the plasma and CSF levels of tryptophan and its downstream metabolites. What is known about the passage of tryptophan and its metabolites across the blood-brain barrier and inflammation? What would a multi-variate analysis utilizing markers of CSF inflammation (or IFN-g as noted by the authors) show? Would tryptophan CSF level still remain as an independent predictor of mortality?

4. It has been shown previously that ventricular and spinal CSF have different profiles (and acknowledged by the authors in the current manuscript). Since spinal CSF was utilized by the current studies, it would be interesting to know how many of the patients in this cohort also had spinal TB and if this was correlated with tryptophan/metabolite levels.

5. The authors suggest the use of animal models, but do not quote any for TB meningitis, but which could be quoted (PMID: 29777209, 32501258, 30518610, 35085105, 36581633).

---

## [Author Response]

Essential revisions:1. Modelling. The impact of baseline CSF and plasma metabolite levels on 60-day survival was tested in a Cox-regression model, adjusted for sex, age, and HIV. One of the research questions could be whether tryptophan concentrations and outcome in TBM is a defining factor for outcome or just simply a bystander effect of damage. To study this in a cohort with >1000 TBM patient and 231 deaths, the authors might want to perform multiple regression analyses for 30-day survival, including baseline and clinical features presented in the baseline table. They could include tryptophan concentrations in categories depending on the distribution of values (i.e., quartiles) and perform multiple imputation for missing values. While the association of CSF tryptophan levels with mortality (high tryptophan levels associated with higher 60-day mortality) are an interesting finding, it is confusing to note that uninfected control subjects had much higher CSF tryptophan levels (Figure 3). Similar, trend to TB meningitis (i.e. higher mortality in those with higher tryptophan levels) were noted in patients with cryptococcal meningitis. Finally, patients with bacterial meningitis also had much higher tryptophan levels but information on mortality is lacking. Therefore, it is likely that CSF tryptophan levels may not be the driver of mortality and more likely is an inflammatory biomarker in chronic forms of meningitis. This needs more detailed analysis and discussion.

We thank the reviewer for commenting on the comparison with control patients, and on causality with mortality within the group of TBM patients.

First, on the comparison with controls patients. We think that the fact that TBM and cryptococcal patients had lower CSF tryptophan than non-infectious controls, points to active tryptophan metabolism (i.e. catabolism) in TBM cryptococcal meningitis, which apparently does not occur in bacterial meningitis patients, who had similar CSF tryptophan levels compared to non-infectious controls.

Then, regarding the multivariable analysis. All the current analyses are multivariable, including age, sex and HIV as covariates, and stratified for study site as the overall mortality is higher in Indonesia than Vietnam. We deliberated the inclusion of disease severity as another covariate in our analysis plan. We think that Glasgow Coma Scale is the most appropriate proxy for disease severity because it predicts outcome in both our cohorts and because of its more granular ordinal scale (3-15) compared to TBM grade (1-3). Tryptophan did not correlate to GCS in the previous paper (van Laarhoven et al., *Lancet Infectious Diseases* 2018). For the analysis plan for the current study, we considered the following options for disease severity (i.e. GCS):

GCS as an intermediate between tryptophan and outcome.

GCS being an independent predictor for outcome.

GCS driving mortality, with tryptophan as an intermediate.

We argued that:

In case 1, adding it to a multivariable model would be problematic because of multicollinearity.

In case 2. adding GCS would not change the result.

Case 3 we thought unlikely, because of the lack of correlation in our previous analysis.

That said, in the current study, GCS had a very week negative correlation with tryptophan (r = -.08), i.e. patients with severe disease (lower GCS, worse outcome) had a slightly higher tryptophan (associated to worse outcome). We can therefore understand the reviewers concern, although the correlation is only weak. And because the correlation is only weak, multicollinearity is not much of a problem. We performed two additional multivariable models:

including the pre-defined variables, and adding GCS.including the pre-defined variables, and adding GCS and the other baseline variables.

We have added to the Results section (lines 211-217) “Because of weak negative correlation between tryptophan and GCS (r = -0.08) and to exclude the possibility that tryptophan is a marker of patients with more severe disease, we performed a post-hoc analysis with two additional models. These models included the basic pre-defined variables (age, sex, HIV as covariates, study site as stratum). Adding GCS did not substantially change the effect size (HR 1.15, 95% CI 1.08-1.23), and adding GCS and CSF cell counts, CSF protein and glucose ratio did neither (1.14, 95% CI 1.07-1.22) confirming that tryptophan was associated to mortality independent of the aforementioned parameters.”

We have also added a sentence at the end of the discussion (lines 355-357) referring to the above and acknowledging that with the current data we cannot infer a definite causation.

“The lower CSF tryptophan values in TBM patients compared to non-infectious controls, implies active tryptophan metabolism in TBM. Combined with an improved survival in TBM patients with the lowest CSF tryptophan, this could imply that in TBM an active tryptophan metabolism is beneficial. Interventional studies will be needed to confirm this hypothesis.”

2. Study power. Based on the previous findings and the number of patients in the cohorts, the authors could say something about the statistical power, i.e., what are changes that they are able to detect.

We performed a power calculation before the study (as part of the grant application): “We based the sample size calculation on the variance for tryptophan metabolites with neuromodulating effect (kynurenic acid, 3-hydroxyanthranilic acid and quinolinic acid) in the original discovery cohort (n = 32). Applying strict Bonferroni-correction for the total number of tryptophan metabolites measured, 96-213 events (non-survivors) are needed in each group for adequate power. For HIV-uninfected patients (estimated 33% mortality), this equals to maximum 800 patients and for HIV-infected patients (estimated 50% mortality) to 600 patients for the least discriminating metabolite.“ (ULTIMATE grant proposal).”

In the final analysis plan, because of the similar metabolite levels between HIV-infected and HIV-uninfected metabolites, we decided to incorporate HIV-status as covariate, rather than a stratum, to ensure enough power for this confirmatory study.

We have added the change in data analysis (lines 173-175) to improve power by incorporating HIV-status in our analyses.

3. Early vs. late mortality. The association between tryptophan concentrations and outcome in TBM seems to be not only true for the early period (e.g., the first 14 days), but also for the late period. As suggested in the results (HIV = vs HIV- patients; suppl table 1). Could the authors explain in more detail how these analyses were performed? It is difficult to understand how base-line metabolites have a strong effect on late outcome (i.e., at day 180). This might even point towards the hypothesis that found metabolites are not the causing factor of death but more just a bystander effect.

We agree with the reviewers that the finding that baseline CSF tryptophan levels predict later mortality is intriguing. However, we know that events occurring early in treatment (e.g. new infarcts or tuberculomas) have a significant effect on long term outcomes, and that intra-cerebral inflammation lasts for many months (as detected by persistent CSF abnormalities). TBM is slow to resolve, despite optimal therapy, which makes these observations less surprising.

First, with regard to the methods. For patients who died during the specified follow-up period (180 days), the median time to death was 14 days. This was used as the cut-off to differentiate early (day 0 – day 14) from late (day 14 – day 180) mortality. In the analysis for early mortality, 1069 patients entered the analysis of which 155 died. In the analysis for late mortality, the remaining patients entered the analysis, of whom 149 died. As the power of Cox regression is dependent on the number of events (deaths), this is equal for both time periods. The results in this analysis, with similar HRs of 1.14 and 1.17 for both time periods, indicate an ongoing increased hazard for mortality for patients with a high baseline CSF tryptophan. This is illustrated by the Kaplan Meier plots (best seen in Supplementary Figure 3) showing that the lines keep separating beyond the first 14 days. We can hypothesize that an intrinsic difference in tryptophan metabolism, that is shown at baseline, is important also for later mortality, but cannot point to the exact mechanism.

We have added more detail in the methods, lines 112-113.

4. CSF tryptophan is inversely correlated with interferon-γ. 92 inflammatory proteins measured in CSF from 176 TBM patients. 13 were inversely associated with tryptophan concentrations, only interferon-γ was validated in Vietnamese samples. What about the other inflammatory proteins?

We choose to attempt to validate the inverse correlation of CSF interferon-γ with CSF tryptophan in the Vietnamese samples because (1) this correlation was the strongest and (2) biologically the most interesting.

We have added the correlation matrix for the 10 measured cytokines in the Vietnamese patients as Supplementary figure 8B, and described the results in lines 279-280.

5. Defining factor in outcome or bystander. In the first paragraph of the discussion the authors overstate the impact of their results. This is a pity because it is not needed. They state that "Collectively, these finding suggest that tryptophan affects outcome from TBM within the brain rather than systemically. Results do not support that tryptophan affects outcome; they support the statement that tryptophan is associated with outcome.

Thank you for pointing this out. We did not want to claim that tryptophan (like a toxin) increased mortality, and modified our wording accordingly based on your suggestion. We think that a mechanism affecting tryptophan *metabolism* (this is now added in lines 307-308), which takes place localised in the central nervous system (rather than systemically), and potentially driven by interferon-γ influences outcome.

Our previous study in Indonesia provides an extra argument for causality showing that patient genetic correlates of CSF tryptophan concentrations in tuberculous meningitis strongly predicted mortality in a separate group of patients (in which no tryptophan was measured).

6. Negative correlation between tryptophan and IFN. The authors state that they find a strong negative correlation between tryptophan and IFN. The correlation found between tryptophan and IFN (which indeed is stronger than the correlation in Indonesia) has a Spearman's rho of 0.30. Correlation coefficients between 0.3 and 0.5 indicate variables which have a low correlation, not a strong correlation. So, it would be more correct to state that "a higher tryptophan showed a statistically significant but low correlation with IFN-γ". Please correct throughout results and discussion. Better not to overstate your results.

We changed this accordingly (line 265, 285, 307). Of note, the Spearman correlation between CSF tryptophan and IFN-γ was moderate, -0.48 in the Indonesian population and -0.45 in the Vietnamese population (line 283).

The correlation between CSF kynurenine and TNF was 0.58 in the Indonesian population and 0.30 in the Vietnamese population. Of note, as shown in Supplementary Figure 8 and as expected, TNF was part of a larger cluster of pro-inflammatory cytokines. These all correlating positively with CSF kynurenine.

7. There is a large variability in CSF tryptophan levels. If a cutoff were defined to predict the mortality risk, what would be PPV and NPV of this test?

We performed this study to improve our understanding of pathophysiology. Like the clinical variables, CSF tryptophan indeed shows large variation in this cohort.

The study aim was not to identify a prognostic marker and we think our results best support a gradual additive negative effect for higher CSF tryptophan, we therefore did not calculate specific cut-offs.

8. There was some discordance between the plasma and CSF levels of tryptophan and its downstream metabolites. What is known about the passage of tryptophan and its metabolites across the blood-brain barrier and inflammation? What would a multi-variate analysis utilizing markers of CSF inflammation (or IFN-g as noted by the authors) show? Would tryptophan CSF level still remain as an independent predictor of mortality?

We thank the reviewer for these comments. For the first question, to make it clearer, we have added in line 276 “*Tryptophan is transported into the brain by the large neutral amino acid transporters”* (Boado RJ, Li JY, Nagaya M, Zhang C, Pardridge WM. Selective expression of the large neutral amino acid transporter at the blood–brain barrier. *Proc National Acad Sci* 1999; **96**: 12079–84).

As suggested by reviewers, we performed some analyses as below:

1. Including CSF leukocytes, as a proxy for cerebral inflammation, and CSF protein, as a proxy for CSF barrier disfunction, did not change the results (shown under 1.)

2. The cytokine data was incomplete and determined using different methods as indicated. Cytokine data was available for a subset of 178 Indonesian patients. In the multivariable model (including sex, age and HIV status, the HR for tryptophan was 1.13 but with a confidence touching 1 (1.00-1.27) because of a lower number in individuals. Adding IFN-γ to the model did not substantially change the effect of tryptophan, with a HR of 1.13 (0.98-1.30) (Model C). Cytokine data were available for 281 Vietnamese patients. In the multivariable model (including sex, age and HIV status) in this patient subset, the HR for tryptophan was 1.13 (0.99-1.29). Adding IFN-γ to the model slightly decreased the effect of tryptophan to 1.05 (HR 0.91-1.22), Model D).

9. It has been shown previously that ventricular and spinal CSF have different profiles (and acknowledged by the authors in the current manuscript). Since spinal CSF was utilized by the current studies, it would be interesting to know how many of the patients in this cohort also had spinal TB and if this was correlated with tryptophan/metabolite levels.

Spinal MRIs were not usually performed in both cohorts, therefore we cannot identify the proportion with radiologically proven spinal disease. We agree with the reviewer that this would have been of additional value.

10. The authors suggest the use of animal models, but do not quote any for TB meningitis, but which could be quoted (PMID: 29777209, 32501258, 30518610, 35085105, 36581633).

We now completed this statement and included a reference to the animal model for TB meningitis which includes life imaging and could be valuable to study pharmacological interventions aimed at tryptophan metabolism.